# Aerosolized Exposure to H5N1 Influenza Virus Causes Less Severe Disease Than Infection via Combined Intrabronchial, Oral, and Nasal Inoculation in Cynomolgus Macaques

**DOI:** 10.3390/v13020345

**Published:** 2021-02-22

**Authors:** Petra Mooij, Marieke A. Stammes, Daniella Mortier, Zahra Fagrouch, Nikki van Driel, Ernst J. Verschoor, Ivanela Kondova, Willy M. J. M. Bogers, Gerrit Koopman

**Affiliations:** 1Department of Virology, Biomedical Primate Research Centre, Lange Kleiweg 161, 2288 GJ Rijswijk, The Netherlands; mooij@bprc.nl (P.M.); mortier@bprc.nl (D.M.); fagrouch@bprc.nl (Z.F.); verschoor@bprc.nl (E.J.V.); bogers@bprc.nl (W.M.J.M.B.); 2Department of Parasitology, Biomedical Primate Research Centre, Lange Kleiweg 161, 2288 GJ Rijswijk, The Netherlands; stammes@bprc.nl; 3Animal Science Department, Biomedical Primate Research Centre, Lange Kleiweg 161, 2288 GJ Rijswijk, The Netherlands; vandriel@bprc.nl (N.v.D.); kondova@bprc.nl (I.K.)

**Keywords:** influenza, H5N1, infection, aerosol, macaques

## Abstract

Infection with highly pathogenic avian H5N1 influenza virus in humans often leads to severe respiratory disease with high mortality. Experimental infection in non-human primates can provide additional insight into disease pathogenesis. However, such a model should recapitulate the disease symptoms observed in humans, such as pneumonia and inflammatory cytokine response. While previous studies in macaques have demonstrated the occurrence of typical lesions in the lungs early after infection and a high level of immune activation, progression to severe disease and lethality were rarely observed. Here, we evaluated a routinely used combined route of infection via intra-bronchial, oral, and intra-nasal virus inoculation with aerosolized H5N1 exposure, with or without the regular collection of bronchoalveolar lavages early after infection. Both combined route and aerosol exposure resulted in similar levels of virus replication in nose and throat and similar levels of immune activation, cytokine, and chemokine release in the blood. However, while animals exposed to H5N1 by combined-route inoculation developed severe disease with high lethality, aerosolized exposure resulted in less lesions, as measured by consecutive computed tomography and less fever and lethal disease. In conclusion, not virus levels or immune activation, but route of infection determines fatal outcome for highly pathogenic avian H5N1 influenza infection.

## 1. Introduction

Avian influenza viruses naturally circulate in wild aquatic birds but are easily transmitted to poultry. Occasionally, they can infect humans who come into close contact with infected birds, for instance at wild poultry markets or commercial farms [1]. In humans, infection with the highly pathogenic avian influenza virus H5N1 is characterized by a high mortality rate, i.e., 455 deaths in a total of 861 cases [2]. Infection generally starts as a typical upper respiratory tract infection accompanied by clinical symptoms such as fever and body aches. However, it then develops into lower respiratory tract infection and causes a severe pneumonia progressing to acute respiratory distress syndrome and death [3]. At present, it is not clear whether virus replication and local tissue damage or rather the level of immune activation and cytokine production are the main determinants in causing fatal disease. Both an extensive loss of alveolar epithelial cells, leading to perturbation of the alveolar barrier function, and increased levels of proinflammatory cytokines and chemokines in the blood have been described as a hallmark of human H5N1 infection [4,5,6,7].

Non-human primates (NHP) form an important preclinical model in influenza virus research because of their close resemblance to humans with regard to anatomy, physiology, and cellular and humoral immune system, which are well characterized [8,9,10]. Studies in NHP infected with highly pathogenic avian H5N1 virus have contributed importantly to our knowledge on virus localization, pathology, and the stimulation of local and systemic innate and adaptive immune responses [11,12,13,14,15,16]. Similar to humans, NHP infected with H5N1 show an increase in proinflammatory cytokine and chemokine levels in the blood as well as in the lungs [11,13,15]. In addition, virus replication in the nose, throat, and lungs, an extensive loss of alveolar and bronchiolar epithelium early after infection, and the formation of alveolar exudate comprising edema, fibrin, and cell debris [11,12,13,14,15,17] is observed. Typical clinical symptoms are prolonged fever, loss of appetite, and lethargy [14,15,18,19]. However, in contrast to humans, NHP usually recover from the infection, and only occasionally is progression to an acute respiratory distress syndrome observed that necessitates the euthanasia of the animals [14,20].

Experimental infection in NHP is typically performed by either intra-tracheal or a combination of intra-tracheal, oral, intra-nasal and intra-ocular virus inoculation, using a 10^6^ to 10^7^ 50% tissue culture infective dose (TCID_50_) of virus [11,12,13,14,15,17,18,19,20,21,22,23]. However, influenza virus infection in humans is assumed to be mainly caused by exposure to aerosols or droplets that enter the airways either via respiration, inhalation, or via contact with contaminated surfaces [24]. Infection via exposure to aerosols could lead to a different pattern of systemic as well as local immune activation, as was recently shown for pandemic H1N1 (pH1N1) influenza virus [25], and might therefore have an impact on the course of H5N1-induced disease. Aerosolized H5N1 influenza infection was recently studied by Wonderlich et al. and Watanaba et al. [26,27]. However, while Wonderlich et al. described a fulminant pneumonia with a fatal outcome upon H5N1 infection after aerosolized exposure [27], Watanaba et al. did not observe any enhanced disease in animals infected via aerosolized versus so-called “combined-route” exposure [26]. In both studies, highly homologous virus strains were used, i.e., A/Vietnam/1203/2004 (H5N1) versus A/Vietnam/UT/3040/2004 (H5N1) in a comparable dose of 4−5.2 × 10^7^ plaque-forming units (pfu). Both studies were performed in cynomolgus macaques, which have been described to be more susceptible to pH1N1 infection than rhesus macaques [28]. An important difference between these studies may be the collection of bronchoalveolar lavages (BAL), which was performed by Wonderlich et al. at multiple time points during the early phases of the infection [27] and not by Watanaba et al. [26]. Strikingly, in a study described by Rimmelzwaan et al., the one animal that became severely ill had been subjected to BAL collection [14].

In view of these contradictory results regarding the role of aerosolized exposure in causing severe disease in NHP, we evaluated disease severity following H5N1 infection by aerosolized exposure, either with or without BAL collection at day 2, 4, and 7 after virus inoculation and compared the outcome with a combined route of virus inoculation. Furthermore, we aimed to establish whether disease severity was correlated with virus replication and/or changes in immune activation and inflammatory cytokine responses. Consecutive minimal-invasive computed tomography (CT) imaging of the lungs was used to be able to monitor progression of the disease and visualize pathological changes in time, without the need to sacrifice the animals at early time points [25,29]. We observed that combined-route inoculation resulted in higher fever and higher CT scores than aerosol exposure and led to fatal disease in three out of four animals. Collection of BAL from aerosol-exposed animals had no impact on the disease. There was no difference between the groups regarding the level and pattern of virus replication, changes in number or activation of leucocyte subsets, or cytokine and chemokine production in the blood.

## 2. Materials and Methods

### 2.1. Animals 

This study was performed in twelve outbred male, adult, cynomolgus monkeys (*Macaca fascicularis*). Animals were captive-bred for research purposes and socially housed at ABSL-III facilities at the Biomedical Primate Research Center, Rijswijk, Netherlands (an American association for accreditation of laboratory animal care (AAALAC)-accredited institution). Animal housing was according to international guidelines for NHP care and use (The European Council Directive 2010/63/EEC, and Convention ETS 123, including the revised Appendix A as well the ‘Standard for humane care and use of Laboratory Animals by Foreign institutions’ identification number A5539-01, provided by the Department of Health and Human Services of the United States of America’s National Institutes of Health (NIH)). All animal handlings were performed within the Department of Animal Science (ASD) according to Dutch law. The animals were negative for antibodies to simian type D retrovirus and simian T-cell lymphotropic virus, and they were selected for the absence of antibodies directed to influenza A/Vietnam/1203/04 (H5N1) recombinant hemagglutinin (HA) protein (Sino Biological Inc). All animals were classified healthy according to physical examination and the evaluation of complete blood count and serum chemistry.

Animals were pair-housed with a socially compatible cage mate and kept on a 12-h light/dark cycle. The monkeys were offered a daily diet consisting of monkey food pellets, fruit, and vegetables. Enrichment was provided daily in the form of pieces of wood, mirrors, food puzzles, and a variety of other homemade or commercially available enrichment products. Drinking water was available *ad libitum* via an automatic watering system. Veterinary staff provided daily health checks before infection, and the animals were checked for appetite, general behavior, and stool consistency. During the course of the influenza virus infection the animals were checked twice a day and scored for clinical symptoms according to a previously published scoring system (skin and fur abnormalities, posture, eye and nasal discharge, sneezing and coughing, respiration rate) [30]. A numeric score of 35 or more was predetermined to serve as an endpoint and justification for euthanasia. Each time an animal was sedated, the body weight was measured. Body temperature was transmitted continuously via a physioTel Digital Implant (Data Sciences International, (DSI), Harvard Biosciences Inc., Holliston, MA, USA), which was surgically placed in the abdomen of the animal to transmit body temperature in real time. The normal circadian pattern was recorded over a 7-day period before infection and for each 15-min time period of the day, the mean value and 95% confidence interval were calculated. These values (mean plus 95% confidence interval) were subtracted from the temperatures recorded after infection as described previously [28,31]. The temperature increase relative to the normal circadian pattern was calculated for each animal over a 9-day period after virus inoculation. 

### 2.2. Experimental Infection and Influenza Virus Detection

Three groups of four cynomolgus macaques each were exposed to 6 × 10^6^ TCID_50_ of an influenza A/Vietnam/1203/04 (H5N1) virus. Virus was grown on MDCK cells. Virus titer was determined, using a TCID_50_ assay on MDCK cells, as 10^8.64^ (437,000,000) TCID_50_/_mL_. The animals from the combined-route group were inoculated with a suspension containing 10^6^ TCID_50_/_mL_ influenza virus, by intra-bronchial (2 mL left lung, 2 mL right lung, with a bronchoscope), oral (1 mL), and intra-nasal (0.5 mL per nostril) administration, total amount of virus of 6 × 10^6^ TCID_50_. The aerosols were generated using an OMRON U22TM nebulizer (OMRON Healthcare Co Ltd., Kyoto, Japan) with electronic mesh that generates particles with a mass median aerodynamic diameter (MMAD) of 4.2 µm, which can penetrate deep into the respiratory tract [32]. One mL of fluid, containing 6 × 10^6^ TCID_50_ of influenza virus was added to the medication chamber, and nebulization was started. The total amount was 6 × 10^6^ TCID_50_ of influenza virus, the same as in the combined route of exposure. The animals were allowed to inhale the aerosols under general anesthesia (Ketamine hydrochloride 10mg/kg, Alfasan Nederland B.V., Woerden, The Netherlands) via a Laerdal No. 3 child mask (pediatric nose and mouth mask, Laerdal, Wappingers Falls, NY, USA) until all fluid was vaporized, as described previously [25,33]. Aerosol exposure was used in two groups of four animals, one group without and one group with a subsequent collection of BAL fluid on days 2, 4, and 7 after virus exposure. In addition, on day 14 post exposure, BAL fluid was collected from all animals that remained in the study. BAL fluid was collected using a bronchoscope. Tracheal and nasal swabs were collected before and 1, 2, 4, 7, 9, 11, and 14 days post infection using Copan flocked swabs (FLOQswabs, 502CS01, COPAN, Brescia, Italy). Viral RNA was isolated using a QIAamp Viral RNA Mini kit (Qiagen Benelux BV, Venlo, Netherlands) following the manufacturer’s instructions and was detected by real-time PCR as described [34]. After virus exposure, the undiluted as well as diluted virus stocks were back titered on MDCK cells, leading to viral titers of 10^8.9^ TCID_50_/_mL_ for the undiluted stock, 10^6.63^ TCID_50_/_mL_ for the 6 × 10^6^ stock used for aerosolized exposure, and 10^5.67^ TCID_50_/_mL_ for the 1 × 10^6^ stock used for combined-route exposure, which is in agreement with the initially determined titer.

### 2.3. Minimal-Invasive Computed Tomography (CT) Imaging of the Lungs

CTs were performed before and 2, 4, 7, 11, and 14 days after infection to monitor changes in the lungs and visualize pulmonary infiltrates [25]. Animals were sedated with ketamin (10 mg/kg) and combined with medetomidine hydrochoride, 0.05 mg/kg (Sedastart, ASTFarma B.V., Oudewater, The Netherlands) to induce further sedation and muscle relaxation. The animals were positioned in dorsal recumbency. Local anesthesia in the throat was given by spraying with 0.1% Xylocain (Aspen Pharma Trading Ltd., Dublin, Ireland), and the animal was intubated. Animals were placed under isoflurane sedation (1.5%) during CT procedures. Images were obtained using a preclinical MultiScan LFER 150 PET-CT scanner (Mediso Medical Imagings Systems Inc., Budapest, Hungary) with 320 µm slice thickness. All CTs were acquired with a breath-hold of 30 s at a positive end-expiratory pressure (PEEP) of 4 cm H_2_O to eliminate breathing artefacts. CTs were acquired using a semicircular single field of view (FOV) scan method, with an exposure of 90 ms and 1:4 binning using 75kVp, 980µA CT tube strength. CTs were reconstructed with a voxel size of 500 and 1000 µm. CT image analyses was performed using Vivoquant version 4.5 (InVicro, Boston, MA, USA). At the end of the procedure, atipamezol hydrochloride, 0.5 mg/kg (sedastop, ASTFarma B.V., Oudewater, The Netherlands) was used for faster recovery of the animals. The CTs of the lungs were evaluated blindly by two imaging experts and scored using a semiquantitative scoring system as described for coronavirus disease [35,36,37]. In brief, each of seven lung lobes were scored as follows: grade 0, normal image; grade 1, less than 5% of area affected; grade 2, 5–24% affected; grade 3, 25–49% affected; grade 4, 50–74% affected; grade 5, more than 75% affected. A total score was calculated for each animal for each CT scan by adding the score for each lobe, leading to a total maximum score of 35 per time point.

### 2.4. FACS Analysis

Multiparameter fluorescence activated cell sorting (FACS) analysis was performed on ethylenediaminetetraacetic acid (EDTA) blood, using an extensive monoclonal antibody (mAb) panel to measure changes in lymphocyte subset composition, T-cell memory/naïve subsets, and activation in blood, as described previously [28]. The following mAb combinations were used: (a) CD8^V500^, CD16^BV605^, CD20^BV711^, CD45^FITC^, CD159a^PE^, CD14^PE-TxRed^, HLA-DR^PerCP^, CD4^PE-CY7^, CD56^APC^, CD3^AF700^; b) CD95^BV421^, CD8^V500^, CD197^BV605^, Ki67^FITC^, CD25^PE^, CD28^PE-TxRed^, CD45RA^biot/strept-PerCP^, CD4^PE-CY7^, CD103^APC^, and CD3^AF700^. Polystyrene fluorospheres (Beckman & Coulter, Brea, CA, USA) were used to calculate absolute lymphocyte count. These were added to tube A of the EDTA blood samples immediately before lysis. Flow cytometry was performed on an LSRII machine using Diva software (Becton Dickinson).

### 2.5. Assessment of Cytokine and Chemokine Protein Levels in Serum

Cytokine and chemokine concentrations, including interleukin (IL-1β), IL-6, CCL11 (Eotaxin), CXCL10 (IP-10), CXCL11 (I-TAC), CCL2 (MCP-1), CXCL9 (MIG), CCL3 (MIP-1𝛼), CCL4 (MIP-1β), CCL5 (RANTES), CXCL8 (IL-8), TNF𝛼, and IFNγ, were determined using a LEGENDplexTM NHP Chemokine/Cytokine Panel (13-plex) (Biolegend, San Diego, CA, USA) according to the manufacturer’s instruction. Samples were measured on a Aurora (Cytek, Fremont, CA, USA) machine and analyzed by using company software.

### 2.6. Statistical Evaluation

Statistical significance of differences between the groups was calculated by using the unpaired *t*-test. A two-sided α level of 0.05 was used to determine significance. Correlation was calculated by the Pearson correlation test.

## 3. Results

### 3.1. Combined-Route Inoculation and Aerosol Exposure to H5N1 Influenza Virus Results in Comparable Levels of Virus Shedding in Trachea and Nose

In order to evaluate whether aerosolized exposure and collection of BAL could have an impact on the course of H5N1 infection, three groups of four cynomolgus macaques were exposed to 6 × 10^6^ TCID_50_ of the highly pathogenic avian influenza virus strain A/Vietnam/1203/2004 either via intra-bronchial plus oral and intra-nasal inoculation (combined-route), aerosol administration (aerosol), or aerosol administration followed by BAL collection at day 2, 4, and 7 after virus inoculation (aerosol-BAL). The aerosols were generated using an OMRON U22TM electronic mesh that generates particles with a MMAD of 4.2 µm that can penetrate deep into the respiratory tract [32]. All animals became virus positive in the throat (Figure 1A), and there were no significant differences between the groups in the total amount of virus produced in time, which was calculated as area under the curve (AUC) per day to correct for the fact that some animals had to be euthanized before the end of the study (Figure 1C). Instead, virus replication in the nose was only observed in some animals and then at much lower levels than in the throat (Figure 1B). This uneven distribution of virus between the nose and throat was observed in all three groups. Virus was detected in the BAL at day 2, 4, and 7 after virus exposure in all four animals from which BAL had been collected (Figure 1D).

### 3.2. Combined-Route Inoculation with H5N1 Influenza Virus Results in Lethal Disease in Three out of Four Animals and Induces more Lung Lesions and Fever Than Aerosol Exposure

In all animals, irrespective of treatment group, an abnormal or hunched posture, increased breathing, and decreased activity were observed, which peaked at around day 5–8 after infection (Table 1, Figure 2). Some animals showed signs of reduced alertness, but only for one to two days, and this was seen in all groups. Appetite was reduced in almost all animals, but stool or urine were mainly normal. Nevertheless, progression into severe dyspnea was observed in animals C1, C2, and C3 from the combined-route group and animal A1 of the aerosol group (Table 1), while the other animals recovered. Disease was very acute in animal C3, which was found dead in the cage on day 5 after infection. Animals C1 and C2 reached a clinical endpoint and had to be euthanized on day 7 and day 8, respectively. In contrast, only one animal in the aerosol-exposed group (A1) and no animals in the aerosol–BAL group reached a clinical endpoint (Figure 2). Unfortunately, no tissues could be obtained from animal C3. In the other three animals that had to be euthanized, a multi-lobar necrotizing broncho-interstitial pneumonia with multifocal areas of consolidation around or adjacent to bronchi and bronchioles was observed (Figure 3). Loss of respiratory epithelium in the bronchioles and bronchi, extensive alveolar damage accompanied by edema, fibrin deposition, cell debris, and hemorrhage as well as inflammatory infiltrates were seen in all three animals. The cumulative clinical score, averaged per day, was comparable between the groups. However, this result may be skewed by the fact that the animals with the highest disease score had been taken out of study early. Although all animals lost some body weight, this was less than 10% of the starting weight in all animals (not shown).

To monitor the development of lesions in the lungs, a CT analysis was performed on days 2, 4, 7, 11, and 14 after virus inoculation. At day 2 after virus inoculation, lesions were already visible in ten of the twelve animals (Figure 4). By day 4, the affected area had further increased in most animals, resulting in a large proportion of the lungs being occupied by lesions in animals C1 and C3 of the combined-route inoculation group (Figure 5). Subsequently, the CT score decreased by day 11 in the animals that remained in study. The cumulative CT score was significantly higher in the combined-route group than in the aerosol as well as the aerosol–BAL group (*p* = 0.0472 and *p* = 0.0118, respectively). There was no significant difference between the two aerosol exposed groups, indicating that collection of BAL had no effect on the extent of the lesions as measured by CT. The location of the lesions identified by CT was in alignment with the gross pathology findings as determined in the animals that had to be euthanized during the study (not shown). Histopathological examination confirmed that the areas affected represented sites of H5N1-induced pneumonia, as described above.

Temperature was measured continuously and averaged over 15–min by telemetry, using a transmitter that was surgically implanted in the abdominal cavity. First, the normal circadian pattern was calculated for each animal, on the basis of the temperatures recorded during a 7-day period preceding the infection, as described previously [28,31]. Then, these normal values are subtracted from the temperatures that were actually recorded after the infection (Supplemental Appendix A). There was a large increase in temperature observed in all four animals of the combined-route group and one animal from the aerosol exposed group (Figure 4, Appendix A). A clear but more modest increase was seen in the other three animals from the aerosol group and one animal from the aerosol–BAL group. Several animals showed two separate fever peaks, i.e., early after infection from day 2 to 4 and a second peak at day 6 (Supplemental Appendix A), as was also described previously for the pH1N1 infection [28,31]. In the animals that remained in study, the temperature had returned to its normal circadian pattern by day 9. The cumulative temperature increase, i.e., the actually recorded temperature over a 9-day period with normal circadian temperature pattern subtracted was significantly higher in the combined-route group than in the aerosol–BAL group (*p* = 0.0085), while the difference with the aerosol group did not reach the threshold for significance but only showed a trend (*p* = 0.0586). There was no significant difference between the aerosol exposed and aerosol–BAL animals (Figure 4). As previously published for pH1N1 infection in NHP [25], the increase in temperature was positively correlated with the cumulative CT score (r = 0.8591, *p* = 0.003) (Figure 4C).

### 3.3. Combined-Route Inoculation and Aerosol Exposure with H5N1 Influenza Virus Results in Comparable Levels of Acute Lymphopenia, Immune Activation, and Cytokine Release in the Blood

Previous studies have shown that the infection of macaques with H5N1 resulted in an acute lymphopenia and a strong proinflammatory cytokine response in the blood [11,13,15,20,27]. However, we have recently shown that for pH1N1 influenza virus, these responses were only observed in animals that became infected by combined-route exposure and not in animals that became virus positive after aerosol exposure [25]. After infection with H5N1, a transient decrease in lymphocytes was observed on day 1 after virus inoculation in combined-route as well as aerosol-exposed animals (Figure 6A). This decrease involved T-cells (Figure 6A) as well as B-cells and NK cells. Subsequently, these cell numbers started to increase, sometimes rising above the initial cell numbers recorded on the day of infection. Instead, monocyte numbers increased already immediately after infection in all experimental groups, and a strong transient increase in inflammatory CD16-positive monocytes was observed, peaking at day 1 after virus inoculation (Figure 6B). The activation marker CD69 and proliferation marker Ki67, which are increased especially on CD8 T-cells after pH1N1 infection [28], were similarly transiently increased on CD8 T-cells after H5N1 infection in both combined-route as well as aerosol-exposed animals (Figure 6C,D). There was no difference between the aerosol-exposed animals with or without collection of BAL.

Analysis of cytokine and chemokines in serum showed a characteristic peak in IL-6, CCL2, CXCL10, and CXCL11 production at day 1 after virus inoculation (Figure 7) as also described by others for H5N1 and pH1N1 infection [11,15,25,27]. The increase was observed both in the combined-route and aerosol-exposed animals. However, in the animals of the aerosol–BAL group, there was no increase in IL-6 production and the increases in CCL2, CXCL10, and CXCL11 were seen in fewer animals and sometimes at day 2 instead of day 1 after virus inoculation (Figure 7). The difference between the two aerosol groups cannot be explained by a possible effect of BAL collection, since this was performed starting from day 2. However, it must be noted that especially IL-6 is typically increased during a very narrow time interval, peaking at day 1, and it cannot be excluded that in some animals, the peak falls a little earlier or later and is then partially missed on the time of sampling [15]. Previous studies have shown that levels of IL-6 and CCL2 expression can be quite variable [11,15,25,28,38]. There was a striking increase in CCL3 and CCL4 observed in the aerosol–BAL group on day 4, which is two days after the first BAL collection on day 2, that was not observed in the combined-route or aerosol groups and may therefore be induced by the BAL collection procedure (Figure 7). Other proinflammatory cytokines and chemokines, such as IL1-β, TNF𝛼, IFNγ, and CCL5, were not increased except in animal AB3 at one time point (day 4) (not shown). CXCL9 was only increased on day 2 and 4 in animals C4 and C3 (not shown). CXCL8 was highly expressed in all animals before infection; then, it transiently decreased on day 1 to 2, subsequently increased, and then returned to baseline (not shown).

## 4. Discussion

In humans, infection with highly pathogenic avian H5N1 influenza virus often results in severe disease and a high mortality rate [2]. The disease is characterized by a progressive pneumonia with extensive alveolar damage and acute respiratory distress syndrome and is accompanied by the production of high levels of inflammatory cytokines and chemokines [4,5,6,7]. H5N1 infection in NHP shares most of these characteristics, but it differs from infection in humans in that it rarely has a fatal outcome [11,12,13,14,15]. Recently, exposure to H5N1 via the inhalation of aerosols was described to lead to fatal disease [27], while others showed only mild disease after aerosol exposure as well as a combined intra-tracheal, intra-nasal, and intra-ocular virus inoculation [26]. In this study, a combined-route approach, consisting of intra-bronchial, oral, and intra-nasal inoculation was compared with either aerosolized exposure or aerosolized exposure with subsequent collection of BAL on days 2, 4, and 7 after infection. Despite similar levels of virus replication in the throat and nose, immune activation, and cytokine induction in all treatment groups, the combined-route exposed animals developed more severe and fatal disease, with more CT lesions and fever in comparison to the aerosol exposed animals.

Previously, the combined-route inoculation was similarly described to result in comparable levels of virus replication in nose and throat as aerosol exposure in case of pH1N1 influenza virus infection [25]. However, in contrast to the current study, there was no difference in severity, CT score, or temperature increase between the combined-route versus aerosol-exposed animals. These different outcomes might be associated with important differences in receptor distribution; i.e., 𝛼2,6 sialic acid receptors for pH1N1 being predominantly expressed in the upper respiratory tract while 𝛼2,3 sialic acid receptors for H5N1 are mainly found on cells in the lower respiratory tract in humans [39,40]. Hence, direct inoculation in the deep respiratory tract may have more severe consequences for avian than for human influenza viruses. Consistent with these differences in virus receptor distribution, we observed relatively low levels of H5N1 virus replication in the nose, while pH1N1 has been shown to replicate at comparable levels in throat versus nose samples, albeit that high replication in the nose is only seen when intranasal or combined-route exposure is applied [25,41,42,43].

The high proportion of animals developing severe and fatal disease after combined-route exposure in this study contrasts with most of the previous publications [11,12,13,14,15], despite the use of similar highly pathogenic strains and similar virus doses for inoculation. A possible contributing factor might be that here, intra-bronchial instead of intra-tracheal installation was used, which could have led to deeper penetration to the lower lung lobes of the virus. However, in contrast to pH1N1 [25], we did not observe lesions to be preferentially located in the lower lung lobes, as identified by CT analysis. The only study in which a general fulminant pneumonia and fatal outcome was shown after H5N1 infection in macaques was by using aerosolized exposure [27]. In contrast, aerosolized exposure in this study resulted in relatively lower mortality, fever induction, and also fewer CT lesions, especially in direct comparison with combined-route exposure. Both studies show that aerosol exposure leads to a transient lymphopenia, induction of inflammatory monocytes, and high levels of inflammatory cytokine and chemokine production in the blood [27]. Although the collection of BAL was planned to be performed on all animals at day 7 after virus exposure, this was not pursued for the combined-route and aerosol groups because of the severe course of the disease observed in several animals. Therefore, it was not possible to evaluate possible differences in immune activation and cytokine responses in the lungs. Our findings do not support a possible adverse effect of regular collection of BAL early after virus inoculation. Hence, the difference in outcome between our study and the study described by Wonderlich et al. is as yet not fully understood [27]. As discussed previously [25], there is always some level of uncertainty regarding the amount of virus that is actually inhaled and becomes deposited in the lungs via aerosolization. However, the fact that levels of virus replication in the throat were similar between the combined route versus aerosol-exposed animals and that virus was detected at high levels in the BAL in all four animals from which BAL was collected indicates that virus did reach the lungs.

The application of consecutive CT to study influenza virus infection in time has so far only been described in one study in ferrets and in macaques [25,29]. By using longitudinal CTs, we were able to identify distinct lesions in different areas of the lung without the need to sacrifice the animals at early time points and could follow the development of the lesions over time. Analysis of the tissues in three animals that had to be euthanized during the study confirmed that the lesions observed by CT represented typical influenza virus-induced pathological changes in the lungs. Previously, a diffuse inflammation as well as increases in metabolic activity (uptake of Fluor-18 labeled fluorodeoxyglucose (^18^F-FDG) was described, using PET-CT, on day 2 after aerosolized H5N1 exposure in cynomolgus macaques [27], which is consistent with our observations. However, in that study, no subsequent CTs were performed to monitor the further development of the lesions.

We had previously shown that aerosolized pH1N1 influenza infection induced less immune activation and cytokine release in the blood than combined-route inoculation [25]. In contrast, for H5N1, the combined-route and aerosol-exposed animals showed similar transient decreases in lymphocyte, T-cell, B-cell, and NK-cell numbers, acute monocyte activation, and increased Ki67 expression on CD8 T-cells, but not CD4 T-cells, on day 7–10 after infection (Figure 6). Striking was the biphasic increase in CD69 on CD8 T-cells, which was reminiscent of the double fever peak, and the acute peak in Ki67 expression on CD4 and CD8 T-cells that had not been observed upon pH1N1 infection [25,44]. In addition, the induction of proinflammatory cytokines was observed both in the combined-route and the aerosol-exposed animals. Surprisingly, the induction was less prominent or, in the case of Il-6, absent in the animals of the aerosol-BAL group. At present, we cannot fully explain these differences, although the peak expression of IL-6, CCL2, CXCL10, and CXCL11 on day 1 occurs in a rather narrow time frame [11,15,25,28,38], and therefore, the actual peak may have been missed in some animals. In the aerosol–BAL group, there was an increase in chemokine CCL3 and CCL4 expression on day 4, which was not observed in the other groups. Therefore, this may be induced by the BAL collection procedure. However, there was no faster viral clearance, significantly lower CT score, or less fever in this group relative to the aerosol group from which no BAL was collected, and therefore, a possible effect on severity of the disease is not evident.

In this study, a clear distinction was observed between animals exposed to H5N1 by combined-route inoculation becoming severely ill and animals exposed to H5N1 by aerosolized exposure that did develop less severe disease, despite a similar level of virus production in the throat and nose and similar levels of immune activation and cytokine and chemokine release in the blood. Both intra-bronchial and aerosol exposure should lead to penetration of virus deep into the lungs [32]. However, there could be important differences in virus distribution just after virus inoculation. With the combined-route inoculation method, two-thirds of the total injection volume, and hence two-thirds of the virus, is given directly into the bronchi, while it is not known which percentage of the virus ends up in the lungs after aerosol exposure. Further studies may be needed to determine initial differences in virus distribution, for instance by using fluorescently labeled influenza virus, followed by euthanasia and analysis of the tissues. At present, it is unclear whether the use of a bronchoscope might cause some local inflammation that could stimulate local virus and/or cytokine production. Indeed, in the aerosol–BAL group, we observed increased CCL3 and CCL4 production after BAL collection, which however did not lead to a more severe disease. Additional analysis of BAL samples collected early after virus exposure could provide additional answers, but since the procedure itself might also influence the outcome, it may be difficult to fully elucidate the mechanisms involved. Importantly, both the level of virus replication as well as the level of immune activation and cytokine production observed early after infection are not determinate factors for subsequent disease progression.

## Figures and Tables

**Figure 1 viruses-13-00345-f001:**
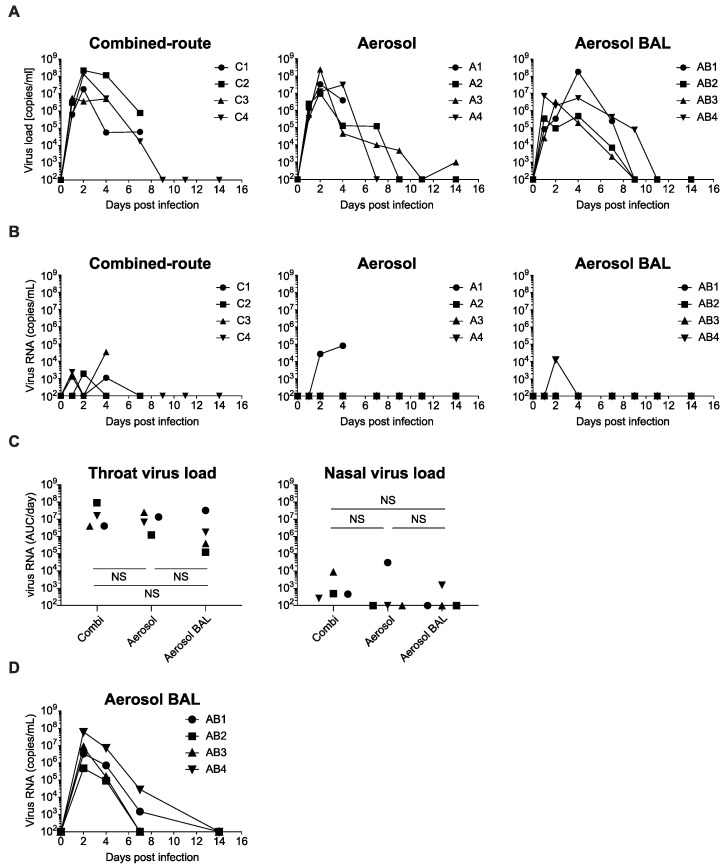
Virus replication in cynomolgus macaques after H5N1 influenza virus exposure. Virus load measured by RT-PCR in time in throat (**A**) and nose swabs (**B**) in animals that received virus by combined-route (left), aerosol (middle), or aerosol delivery with bronchoalveolar lavages (BAL) collection (right). (**C**) Total viral load in throat and nose calculated for each animal as area under the curve (AUC) divided by the number of days that the animal was in study. Symbols used for individual animals correspond with symbols used in line graphs A and B. (**D**) Virus in BAL measured at days 2, 4, and 7 after virus inoculation. Statistical analysis of differences in AUC in throat and nose swabs or in BAL was performed by unpaired *t*-test. NS, not significant.

**Figure 2 viruses-13-00345-f002:**
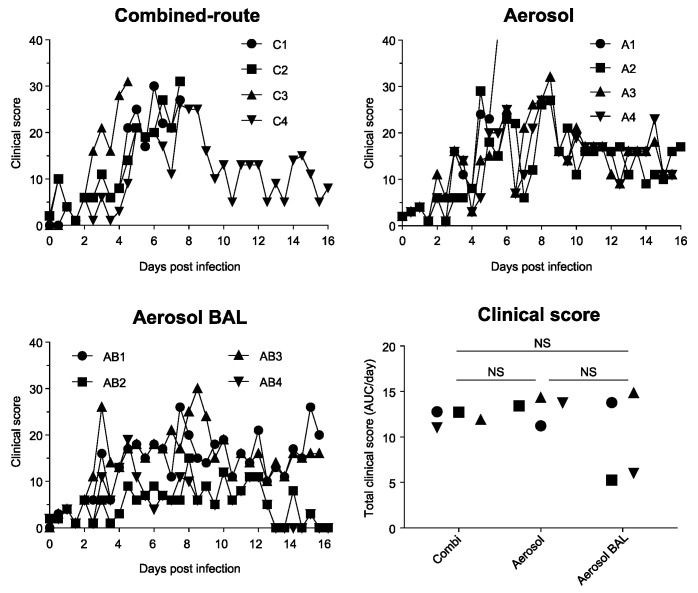
Clinical score in cynomolgus macaques after H5N1 influenza virus exposure. The total clinical score is shown for each individual animal in time after infection with H5N1 by combined-route (upper left), aerosol (upper right), or aerosol delivery with BAL collection (lower left). Total clinical score, calculated for each animal as AUC divided by the number of days that the animal was in study is shown in the lower right graph. Symbols used for individual animals in the scatter plots correspond with symbols used in line graphs. NS, not significant.

**Figure 3 viruses-13-00345-f003:**
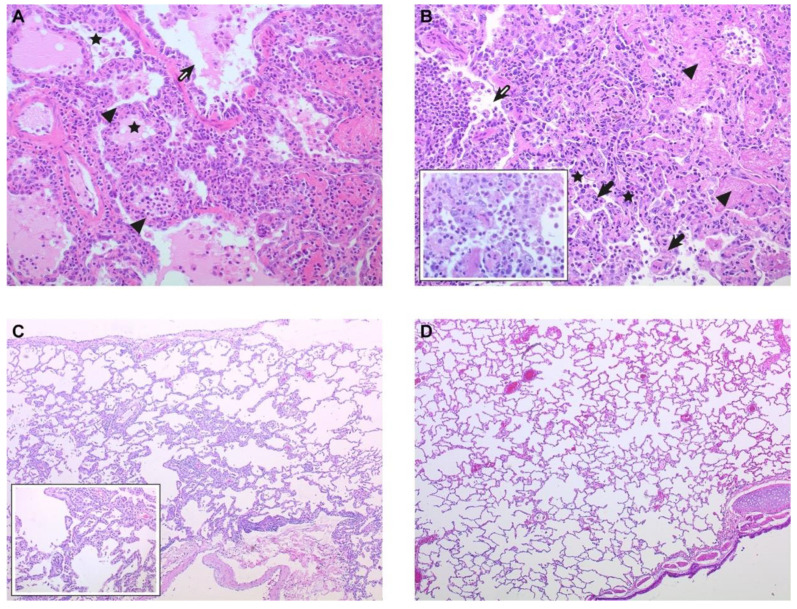
Histologic analysis of the lungs. (**A**) Lung of animal C1 showing broncho-interstitial pneumonia with massive intra-alveolar and intra-bronchiolar edema (hematoxilin–eosin staining (HE) original magnification 200 X). Open arrow, bronchiolar lumen with mixed inflammatory cells and protein-rich edema fluid; asterisk, alveolar lumina with edema and inflammatory cells; closed triangle, alveolar septa expanded by inflammatory infiltrates and edema. (**B**) Lung of animal A1 showing bronchioles and the surrounding alveoli severely obliterated by the influx of mixed inflammatory cells, fibrin, cell debris, erythrocytes, and protein-rich edema fluid (HE, 200 X). Open arrow, bronchiolar lumen filled with mononuclear inflammatory cells and neutrophils; asterisk, alveolar walls expanded by inflammatory cells; closed triangle, alveolar lumina with fibrin and protein-rich edema; closed arrow, alveolar edema with mixed inflammatory cells. Inset: higher magnification of the inflammatory infiltrates (HE, 400 X). (**C**) Lung of animal AB2 17–days post infection showing minimal to mild, focal, resolving lesions (HE, 50X). Inset: higher magnification of focal area with residual chronic and regenerative lesions, including type II pneumocyte hyperplasia, mildly expanded alveolar septa by an increase amount of collagen and infiltration by a small number of macrophages and lymphocytes (HE, 200 X). (**D**) Lung of healthy control macaque (HE, 50 X).

**Figure 4 viruses-13-00345-f004:**
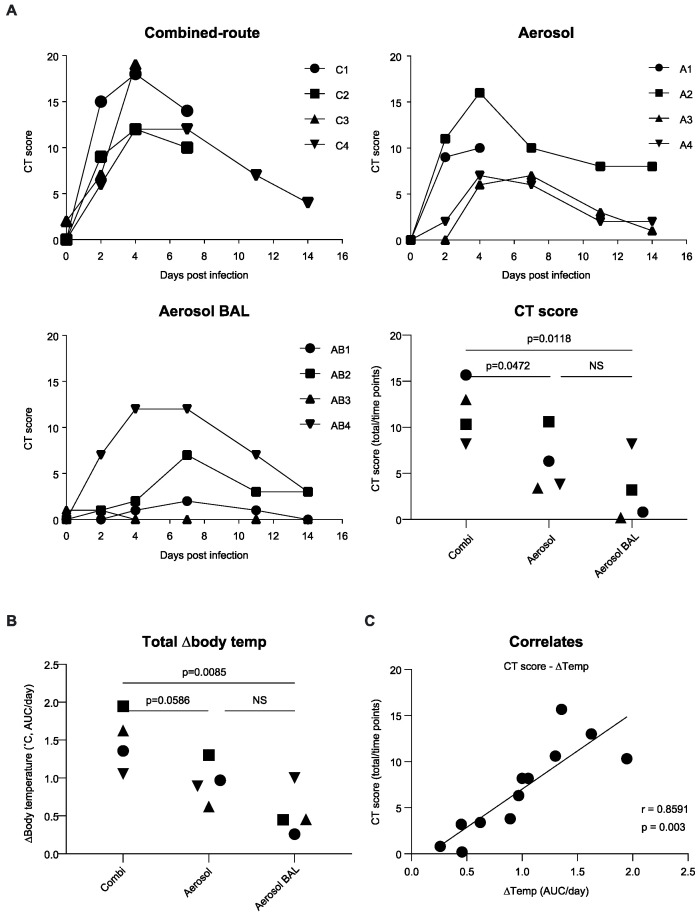
Computed tomography (CT) score and body temperature. (**A**) CT score in time depicted for animals that received virus by combined-route (upper left), aerosol delivery (upper right), or aerosol delivery with BAL collection (lower left). Cumulative CT score, calculated for each animal by adding scores measured at day 2, 4, 7, and 14 and dividing these totals by the number of time points, is shown in the lower right graph. (**B**) Cumulative temperature increase, calculated as the AUC from the actually recorded temperature during 9 days after virus inoculation, minus the circadian temperature pattern recorded before virus exposure, and divided by the number of days that the animal was in study is shown for each individual animal. (**C**) CT score plotted against temperature. Symbols used for individual animals in the scatter plots correspond with symbols used in line graphs. Statistical analysis of differences in AUC was performed by unpaired *t*-test. NS, not significant. The correlation was calculated by Pearson correlation test. The black line represents interpolated data, as a linear curve.

**Figure 5 viruses-13-00345-f005:**
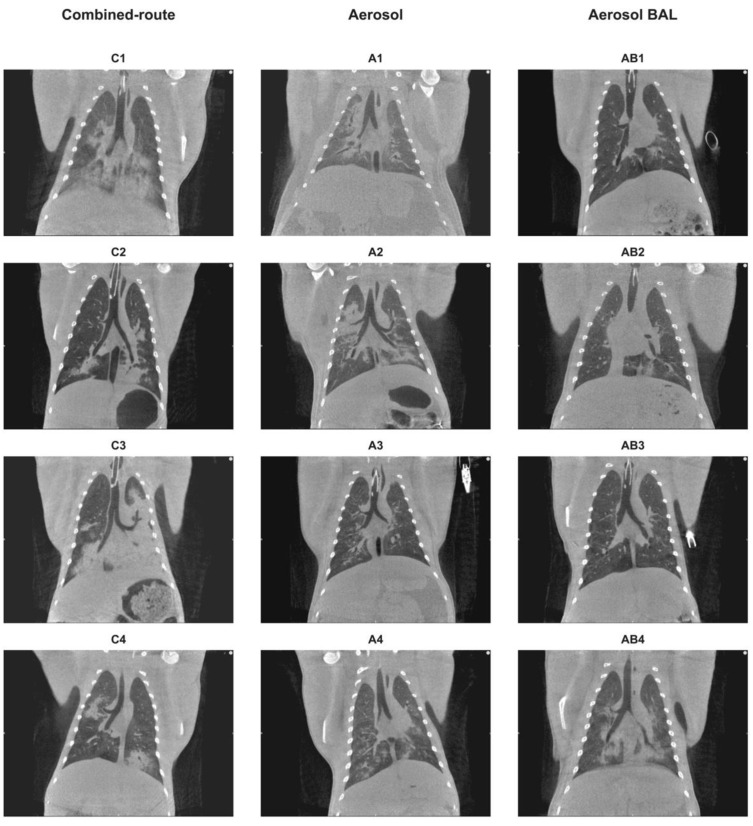
CT images taken on day 4 after virus exposure. Shown are central coronal slices of the lungs on day 4 after virus inoculation of the animals that received virus by combined-route (**left row**), aerosol (**middle row**), or aerosol delivery with regular BAL collection (**right row**).

**Figure 6 viruses-13-00345-f006:**
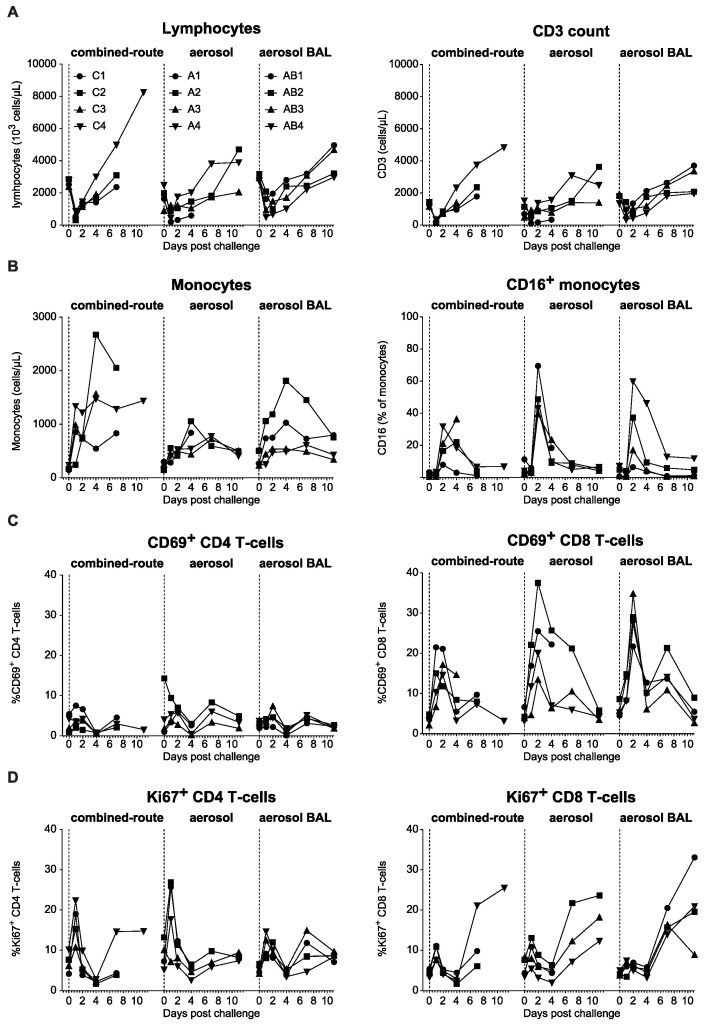
Changes in leukocyte subsets and activation in peripheral blood. (**A**) Peripheral blood lymphocyte and CD3 T-cell count, (**B**) monocyte count and percentage of monocytes expressing CD16, (**C**) percentage of CD4 or CD8 T-cells expressing CD69, and (**D**) percentage of CD4 or CD8 T-cells expressing Ki67 is shown for each individual animal in time for the animals that received combined-route virus inoculation (left side of each graph), aerosol delivery (middle of each graph), or aerosol delivery with BAL collection (right side of each graph). Individual animal numbers are the same for all graphs and only indicated in the upper left graph.

**Figure 7 viruses-13-00345-f007:**
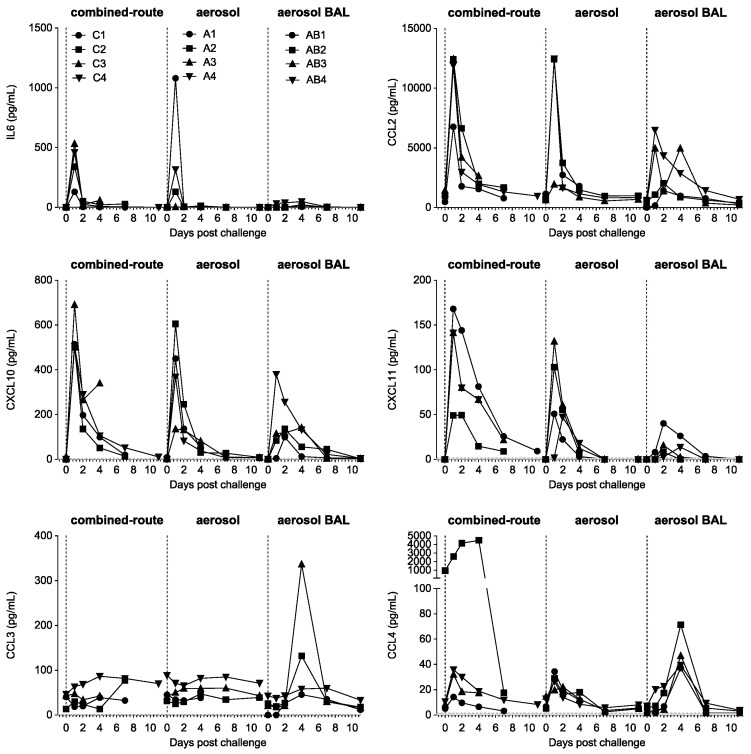
Cytokine and chemokine expression levels in serum. Shown are IL-6, CXCL10, CCL2, CXCL11, CCL3 and CCL4 in pg/mL in serum, for each individual animal in time for the animals that received combined-route virus inoculation (left side of each graph), aerosol delivery (middle of each graph), or aerosol delivery with BAL collection (right side of each graph). Individual animal numbers are the same for all graphs and only indicated in the upper left graph.

**Table 1 viruses-13-00345-t001:** Clinical observations in H5N1 virus-infected animals.

Treatment	Animal ID	Clinical Symptoms ^1^
Combined route	C1	Abnormal posture and increased breathing from day 4 onwards, progressing to severe dyspnea by day 6. Unkempt fur on day 5, reduced activity on day 5. Euthanized on day 7.
	C2	Abnormal posture and increased breathing from day 4 onwards, progressing to severe dyspnea by day 6. Reduced activity from day 5 onwards. Reduced alertness day 7. Euthanized on day 8.
	C3	Severe dyspnea from day 3 onwards. Abnormal posture and reduced activity on day 4. Found dead on day 5.
	C4	Abnormal posture, increased breathing and reduced activity from day 5 to 8.
Aerosol	A1	Abnormal posture, increased breathing, reduced activity, skin redness, reddened eyes on day 5. Euthanized on day 5.
	A2	Abnormal posture from day 4 until day 9. Increased breathing from day 4 onwards. Reduced activity from day 10 onwards.
	A3	Increased breathing from day 2 until day 11. Abnormal posture on day 6, 7, and 8. Reduced alertness on day 6. Reduced activity from day 7 onwards.
	A4	Increased breathing and reduced activity from day 5 onwards. Abnormal posture on day 6 and 7.
Aerosol BAL	AB1	Abnormal posture, increased breathing and reduced activity from day 3 onwards. Reduced alertness on day 15.
	AB2	Increased breathing from day 6 to 12.
	AB3	Abnormal posture from day 2 to 8. Increased breathing and reduced activity from day 4 onwards. Reduced alertness on day 7 and 8.
	AB4	Increased breathing from day 3 to 12. Abnormal posture or reduced activity observed on day 4, 5, and 7 and on day 4 and 11, respectively.

^1^ Animals were monitored twice daily during the course of the infection and scored for clinical symptoms according to a previously published scoring system; skin and fur abnormalities, posture, eye and nasal discharge, sneezing and coughing, respiration rate [30].

## Data Availability

The data presented in this study are available on request from the corresponding author. The data will be made public upon publication of the manuscript.

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
