# Peer review of "Aerosolized Exposure to H5N1 Influenza Virus Causes Less Severe Disease Than Infection via Combined Intrabronchial, Oral, and Nasal Inoculation in Cynomolgus Macaques"

_viruses, 2021, doi:10.3390/v13020345_

Round 1
Reviewer 1 Report
Mooij et al. demonstrated that aerosolized exposure to H5N1 influenza virus causes less severe disease than infection via combined intrabronchial, oral and nasal inoculation in monkey model. They also suggested that not virus levels or immune activation, but route of infection determines fatal outcome for highly pathogenic avian H5N1 influenza infection. Although it still remains unknown why the phenomenon was shown, the fact that they demonstrated is very important to proceed the future pathogenic research for animal model of H5N1 influenza virus infection.
Major concerns:
The authors suggest that not virus levels or immune activation, but route of infection determines fatal outcome for highly pathogenic avian H5N1 influenza infection; however, they have not shown the comparison of virus distribution between by aerosolized exposure and combined inoculation. If they determine the distribution of virus with a virus labeled with a fluorescent or luminescent dye, they will be able to discuss a little bit clearly why the route of infection determines fatal outcome for highly pathogenic avian H5N1 influenza infection.
Maybe combined inoculation with avian H5N1 influenza virus infects alveolar epithelial cells directly more frequent than aerosolized exposure. Thus, the reviewer predicts that combined inoculation with avian H5N1 influenza virus induces severe lung inflammation because only the monkeys with combined inoculation may have influenza virus in lungs just after the inoculation. Authors should explain and discuss whether the reviewer’s prediction is completely wrong or something wrong, and should explain or discuss why route of infection determines fatal outcome for highly pathogenic avian H5N1 influenza infection.
Minor concerns:
Superscript and subscript numbers are not listed. For example, Line 138; 6x106 TCID50 should be changed to 6x106 TCID50. There are many other parts that need to be fixed in the same way.
The location of the company name described in the description of the reagents and equipment described in the method is missing. For example, Line 144; Is OMRON company name? If yes, where is this company?
Line 219; what is means “vaccine group”?
Author Response
Reply to comments reviewer 1
Below a point to point reply is given to the issues raised by the reviewer. Where changes were made to the text of the manuscript this is specified in italics.
The reviewer correctly states that we did not study virus distribution directly after virus exposure. However, we did show that aerosol exposure resulted in high amounts of virus being produced already two days after exposure. Therefore, it is not likely that only monkeys with combined inoculation had influenza virus in the lungs just after the inoculation. Nonetheless, measuring initial virus distribution by using fluorescently labelled influenza virus could indeed help to better understand initial differences in distribution and provide a possible explanation for the observed differences in pathogenicity. However, this would require a different study setup as animals need to be sacrificed immediately after virus inoculation. Other factors like an inflammatory effect of injection of fluid into the lungs could also have contributed to enhanced virus replication. Collection of BAL on day 2 in the combined-route inoculated animals could provide further information about early levels of virus replication, but was not applied here since its effects were not yet known.We have modified the discussion (line 579-585) to address this issue and elaborate more on the possible causes for a more severe disease outcome. The following text was added: “However, there could be important differences in virus distribution just after virus inoculation. With the combined-route inoculation method two-thirds of the total injection volume, and hence tow thirds of the virus, is given directly into the bronchi, while it is not known which percentage of the virus ends up in the lungs after aerosol exposure. Further studies may be needed to determine initial differences in virus distribution for instance by using fluorescently labeled influenza virus, followed by euthanasia and analysis of the tissues”.
We corrected the mistakes in superscript and subscript numbers. Line 71, 146, 149, 151 etc.
Names of companies were added, for instance at line 135 etc.
In line 219 (now 277)
“vaccine group” was a mistake. It is now removed.
Reviewer 2 Report
The manuscript is well-written. Methods are appropriate and well described. The data presented is valid and robust. I have a few comments for the authors to address.
In the discussion please address the reason why the combined route would cause a more severe disease outcome. Since the virus should reach the lungs via the aerosol route as well, what would be a possible explanation for the difference seen? Would this be explained by the larger volume of the inoculum used for the combined-route group (5.5 ml vs. 1 ml for aerosol groups)? If that's the case, then the combined-route group received an inoculum with a higher viral load than the aerosol groups.
Was the back titer of the inoculum determined after the inoculation of the animals? If so, please add this information to the text.
Also, please add histopathology pictures showing differences in pulmonary lesions between groups.
I also have a few minor editorial changes - please see attached file.
Author Response
Reply to comments reviewer 2
Below a point to point reply is given to the issues raised by the reviewer. Where changes were made to the text of the manuscript this is specified in italics.
At the moment we do indeed not fully understand what is causing the difference in pathogenicity upon combined-route versus aerosolized influenza virus exposure. As is stated in the discussion, we do not see a difference in virus replication in the nose or throat or differences in levels of immune activation or proinflammatory cytokine responses. It could be that with the combined-route a relatively larger proportion of the virus dose ends up in the lungs, since as the reviewer correctly states, 4 ml of the total of 6 ml inoculation volume is given directly in the bronchi. For the aerosolized exposure we do not know which percentage of the virus ends up in the lungs. Further studies may be needed to determine initial differences in virus distribution for instance by injecting fluorescently labeled influenza virus followed by immediate euthanasia and analysis of the tissues. On the other side we did observe high levels of virus replication in the BAL at day 2 after aerosol exposure, showing that the lungs were adequately reached. Nonetheless, we do not know whether in the animals receiving the combined-route inoculation virus levels at day 2 would even be higher. Collection of BAL on day 2 in the combined-route inoculated animals could provide further information about early levels of virus replication, but was not applied here since at the start of the study it was not yet known how BAL collection would affect virus replication and disease progression. We have modified the discussion (line 579-585) to address this issue and elaborate more on the possible causes for a more severe disease outcome. The following text was added: “However, there could be important differences in virus distribution just after virus inoculation. With the combined-route inoculation method two-thirds of the total injection volume, and hence two-thirds of the virus, is given directly into the bronchi, while it is not known which percentage of the virus ends up in the lungs after aerosol exposure. Further studies may be needed to determine initial differences in virus distribution for instance by using fluorescently labeled influenza virus, followed by euthanasia and analysis of the tissues”.
The combined-route group received in total 6 ml of a 106TCID50solution, while the aerosol groups received 1 ml of a 6x106TCID50solution. In total the same amount of virus was used for the combined-route and aerosol exposed groups.The description of the amount of virus given is indeed somewhat confusing. Therefore, at line 157-158 the following sentence was added: “The total amount was 6x106TCID50 of influenza virus, the same as in the combined-route of exposure.”
The back titer of the inoculum was indeed determined.The following information is added at lines 170-174: After virus exposure the undiluted as well as diluted virus stocks were back titered on MDCK cells, leading to viral titers of 108.9 TCID50/mlfor the undiluted stock,106.63 TCID50/ml for the 6 x 106stock used for aerosolized exposure and 105.67 TCID50/mlfor the 1 x 106stock used for combined-route exposure, in agreement with the initially determined titer.
An extra figure (now figure 3) was added showing typical lesions in an animal with severe disease from the combined-route and the aerosol group and mild lesions in an animal from the aerosol-BAL group. It should be noted that the animal from the aerosol-BAL group was kept in study until the end and euthanized at day 17 after virus exposure, while C1 from the combined-route group was euthanized at day 7 and animal A1 from the aerosol group at day 5.
Round 2
Reviewer 1 Report
The revised manuscript takes into account what I have pointed out and provides a sufficient reaction and consideration to it. Therefore, the revised manuscript can be accepted by this journal.